# Novel Cytoplasmic Bacteriocin Compounds Derived from *Staphylococcus epidermidis* Selectively Kill *Staphylococcus aureus*, Including Methicillin-Resistant *Staphylococcus aureus* (MRSA)

**DOI:** 10.3390/pathogens9020087

**Published:** 2020-01-30

**Authors:** In-Taek Jang, Miso Yang, Hwa-Jung Kim, Jeong-Kyu Park

**Affiliations:** 1Department of Microbiology and Department of Medical Science, College of Medicine, Chungnam National University, Daejeon 35015, Korea; janginteak@naver.com (I.-T.J.); iammiso@gmail.com (M.Y.); hjukim@cnu.ac.kr (H.-J.K.); 2Cancer Research Institute, College of Medicine, Chungnam National University, Daejeon 35015, Korea

**Keywords:** bacteriocin, *Staphylococcus epidermidis*, *Staphylococcus aureus*, TCA/acetone precipitation method

## Abstract

*Staphylococcus aureus* (*S. aureus*) is one of the well-known agents causing atopic dermatitis (AD) in susceptible individuals, and *Staphylococcus epidermidis* (*S. epidermidis*) produces class I thermostable bacteriocins that can selectively kill *S. aureus*, suggesting protective roles against AD. There is a large need for developing precise therapies only to target *S. aureus* and not to harm the beneficial microbiome. On the agar well diffusion assay, live planktonic *S. epidermidis* showed clear zones of inhibition of *S. aureus* growth, but heat-killed cells and cell-free supernatants did not show this. These results would lead us to hypothesize that cytoplasmic bacteriocin from *S. epidermidis* will be a promising agent to inhibit *S. aureus* growth. Therefore, we have extracted a novel thermolabile cytoplasmic bacteriocin from *S. epidermidis* using trichloroactic acid (TCA)/acetone precipitation method after cell lysis with a SDS-containing buffer. These bacteriocin selectively exhibited antimicrobial activity against *S. aureus* and methicillin-resistance *Staphylococcus aureus* (MRSA), presenting no active actions against *S. epidermidis*, *E. coli*, and *Salmonella* Typhimurium. The extracted cytoplasmic bacteriocin compounds revealed several diffuse bands of approximately 40–70 kDa by SDS-PAGE. These findings suggest that these cytoplasmic bacteriocin compounds would be a great potential means for *S. aureus* growth inhibition and topical AD treatment.

## 1. Introduction

Atopic dermatitis (AD) is a chronic inflammatory skin disease that manifests as dry skin and eczematous dermatitis with prominent itch [1]. Despite its complexity, pathogenesis of AD is quite obviously associated with the skin microbiome. Recently, the correlation of *Staphylococcus aureus* (*S. aureus*) with AD during active disease exacerbation has been well documented [2,3,4,5]. AD is a long-standing inflammatory skin disease typified by epidermal barrier dysfunction that can affect the bacterial community of the skin. A *S. aureus* cell wall product, lipoteichoic acid (LTA), is shown to cause skin barrier damage by inhibiting the expression of epidermal barrier proteins filaggrin and loricrin. Dysbiosis contributes to the pathogenesis of AD by both detrimental effects from *S. aureus* and by loss of beneficial effects from some members of the microbiome. *S. aureus* secretes a pore-forming, phenol soluble modulins (PSMs, δ-toxin) which destroys the skin barrier and promotes skin inflammation as a fuel for dermal mast cells. AD patients frequently demonstrate increased levels of IgE and production of IgE specific for staphylococcal superantigens whose levels are positively correlated with AD severity [6,7]. In our previous study, we found that live planktonic *S. aureus* or methicillin-resistant *S. aureus* (MRSA) induced human mast cell degranulation, but no effect had been found in heat-killed bacteria [8].

Current treatments for AD with the usage of topical steroids improve dermatitis due to their strong anti-inflammatory effect. However, skin thinning with long-term use and recurrence of dermatitis after discontinuation are highly associated [9]. In order to compensate for these limitations in those conventional topical steroid treatments, several attempts have been made to achieve the removal of colonized *S. aureus*. However, studies imply that topical antibiotic therapy would not be able to reduce bacteria counts and theoretically protect the normal skin microbiome [10]. Furthermore, the frequent use of pharmaceutical antibiotics promotes antibiotic resistance [11]. Consequently, there is a rising need again to develop pathogen-specific precision therapies only to target the infectious pathogens and not to harm the beneficial microbiome.

Multiple studies are currently focused on the manipulation of the skin microbiome to explore its therapeutic potential. The colonizing bacteria (*S. epidermidis*) are probably in strong competition and may use a variety of strategies to overcome competitors (*S. aureus*). Bacteria from human microbiota have occasionally been found producing bacteriocins, antimicrobial substances that show an activity against closely related bacteria [12]. Bacteriocins produced by *S. epidermidis* have bactericidal effects towards *S. aureus*. These antimicrobial peptide-producing coagulase-negative *Staphylococcus* (CoNS) strains are less frequent in atopic individuals, and the reintroduction of CoNS decreases *S. aureus* colonization. It has been reported that transplant of *S. hominis* and *S. epidermidis* strains secreting antimicrobial peptides (AMPs) is effective in controlling *S. aureus* overgrowth [7,13]. However, *S. epidermidis* is an opportunistic pathogen holding an emerging risk factor in hospitals worldwide and is often difficult to eradicate due to its virulent strains producing a protective biofilm matrix [14,15]. Therefore, single *S. aureus* targeting bacteriocin can become a valuable strategy for preventing *S. aureus* colonization.

*S. epidermidis* ATCC12228 is an avirulent, non-infection-associated and non-biofilm-forming strain. The bacteriocins produced by this *S. epidermidis* strains are low-molecular-weight (lower than 5 kDa) and heat-stable class I lantibiotics. The low-molecular-weight (< 100 kDa) proteins are not observed in cell-free supernatant (CFS) from *S. epidermidis* ATCC12228. The majority of the identified proteins (~80%) were predicted to be cytoplasmic in this strain. This indicates that these proteins are expressed but not exported by *S. epidermidis* ATCC12228 [16,17]. We also observed that live planktonic *S. epidermidis* ATCC12228 generated a clear zone of growth inhibition *S. aureus* ATCC 25923, but heat-killed cells and CFS did not show any activities on the agar well diffusion assay. These results revealed that this strain itself could not spontaneously secrete bacteriocins into the CFS at a level of sufficiency directly to inhibit *S. aureus*.

Thus, the aim of this study was to extract cytoplasmic bacteriocin compounds from *S. epidermidis* ATCC12228 by TCA/acetone precipitation method after cell lysis with SDS-containing buffer. We suggest that the application of cytoplasmic bacteriocin compounds in this strain is a safer and more effective therapeutic method than those of live *S. epidermidis* for the growth inhibition of *S. aureus* in AD. Herein, we describe the partial purification and characterization of the novel thermolabile cytoplasmic bacteriocin compounds from cell extract of the *S. epidermidis* strain. These bacteriocin compounds from *S. epidermidis* are a novel bacteriocin that have potential applications in management of AD.

## 2. Results

### 2.1. Antimicrobial Activity of Live Planktonic S. epidermidis or CFS from S. epidermidis

*S. epidermidis* (ATCC12228 and NCCP14768) were screened for antimicrobial activity against *S. aureus* (ATCC25923, NCCP14780 and MRSA), *E. coli* and *Salmonella* Typhimurium by agar well diffusion assay. Both live planktonic *S. epidermidis* strains showed antimicrobial activity against all *S. aureus* strains displaying clear zones as a result of inhibition, but heat-killed cells did not show any activities [Figure 1A]. None of the *S. epidermidis* strains observed inhibition zones for all *S. aureus* strains with the treatment of CFS only [Figure 1B]. The inhibition ability of live planktonic *S. epidermidis* was detected against *S. aureus* ATCC25923 (11.1 ± 0.1 mm), *S. aureus* NCCP14780 (11.4 ± 0.2 mm) and MRSA ATCC33591 (10.6 ± 0.1 mm). No inhibition activity was observed in *S. epidermidis*, *E. coli* and *Salmonella* Typhimurium with the combined treatment of both live planktonic strains and CFS [Table 1]. The results suggested that the live planktonic *S. epidermidis* selectively kill *S. aureus*, including MRSA.

The average pH of skin surfaces lies between 5.0 and 6.0. The effect of pH in culture medium on the production of the cytoplasmic bacteriocin compounds has been assessed. The inhibition zone of intra-cytoplasmic protein (IP) from *S. epidermidis* ATCC12228 at pH 5.0 (10.8 ± 0.1 mm) or 6.0 (10.4 ± 0.2 mm) present more prominent actions than that of pH 7.4 (8.6 ± 0.1 mm) against *S. aureus* or MRSA. No antimicrobial activity was observed on the CFS with overnight growth culture of *S. epidermidis* (Figure 2).

### 2.2. Antimicrobial Activity of Concentrated Proteins from CFS of S. epidermidis Strains

As *S. epidermidis* grown on the agar well diffusion assay generated a clear zone of inhibition of *S. aureus* growth but not in CFS, it is necessary to isolate sufficient amounts of bacteriocin for the antimicrobial activity. Thus, we precipitated CFS by 10% TCA and precipitated proteins were re-suspended with solubilization solution. The precipitated proteins of CFS did not show any antimicrobial activity against all *S. aureus* on agar well diffusion assay, but the reference strain of *Bacillus subtilis* showed the antimicrobial activities toward *S. aureus* (data not shown).

### 2.3. Antimicrobial Activity of Cytoplasmic Bacteriocin Compounds from S. epidermidis Strains

Inquiries came from where live planktonic *S. epidermidis* only have the antimicrobial activity against all *S. aureus*, but not CFS or precipitate proteins of CFS on the agar well diffusion assay. To answer this question, we performed the extraction of cytoplasmic proteins from dried *S. epidermidis* by TCA/acetone precipitation method after lysis by SDS-containing buffer.

The extracted cytoplasmic bacteriocin compounds were tested for antimicrobial activity against *S. aureus* by agar well diffusion method. The production of a clear zone by growth inhibition was noted. The antimicrobial activity of cytoplasmic bacteriocin compounds were found to be stable after heating at 45 °C for 20 min. However, it seemed that it lost its antimicrobial activities on heating to 100 °C for 20 min (Figure 3).

### 2.4. The Minimum Bactericidal Concentration of Cytoplasmic Bacteriocin Compounds

On the agar well inhibition assay, the concentration ranges of cytoplasmic bacteriocin from 2 mg, 1 mg, 500 μg, 250 μg, and 125 μg/ ml in each well and the diameter of growth inhibition area was measured as 14.4 ± 0.2, 12.1 ± 0.1, 10.2 ± 0.2 mm, 8.3 ± 0.1 mm, and none, respectively [Figure 4]. Based on our experiments, 250 μg/ml concentration yields 10.2 ± 0.2 mm the diameter of the growth inhibition zone. We conducted the various concentration ranges from above/below the 250 μg/ml concentration. Antimicrobial activity of cytoplasmic compounds from *S. epidermidis* ATCC12228 at a concentration of 125 μg/ml did not exhibit a growth inhibitory effect on *S. aureus* ATCC25923. Therefore, the minimum bactericidal concentration of cytoplasmic bacteriocin compounds was 250 μg/ml.

### 2.5. The Molecular Weight Ranges of Partially Purified Cytoplasmic Bacteriocin Compounds

The molecular weight of partially purified cytoplasmic bacteriocin compounds was determined by SDS-PAGE gel electrophoresis method. The examination of the cytoplasmic bacteriocin compound profiles by SDS-PAGE exposed several diffuse bands of approximately 40–70 kDa [Figure 5].

## 3. Discussion

In this present study, the live planktonic *S. epidermidis* ATCC12228 showed a specific antimicrobial activity targeting only *S. aureus* strains with clear zones of inhibition on the agar well diffusion assay, whereas heat-killed *S. epidermidis* showed no growth inhibition areas [Figure 1]. No inhibitory actions were observed in *S. epidermidis*, *E. coli* and *Salmonella* Typhimurium with both live planktonic strains and CFS [Table 1]. The results suggested that the only live planktonic *S. epidermidis* selectively kill *S. aureus*, including MRSA.

Dysbiosis contributes to the pathogenesis of AD in both detrimental effects from *S. aureus* and loss of beneficial effects from some members of the microbiome. Pharmaceutical antibiotics have been used to inhibit *S. aureus* in the management of AD, but their efficacy on the skin is limited and has disadvantages in that they may kill beneficial strains and break mutualistic interactions between the skin and microbial communities. High potential antimicrobials may result in a short-term improvement, but they can subsequently increase a long-term risk by causing dysbiosis and promoting antibiotic resistance [7,13,18].

Human skin is highly tolerant to exposure to *S. epidermidis*, which may contribute to the colonization of human skin by *S. epidermidis* as a permanent commensal. Although the role of *S. epidermidis* in skin biology is still emerging, there is increasing evidence that *S. epidermidis* plays an important role in skin defense especially by directly restricting the growth of pathogenic bacteria such as *S. aureus* and activation of immune cells. CoNS are typically low virulence organisms that commonly colonize the skin and mucous membranes. CoNS, including *S. epidermidis*, compete with *S. aureus* for stable colonization of human skin reservoirs by production of antimicrobial factors [7,9,19].

Live planktonic *S. epidermidis* have the antimicrobial activity only against all *S. aureus*, but not CFS or precipitated proteins of CFS on the agar well diffusion assay. Low-molecular-weight (< 100 kDa) proteins from *S. epidermidis* ATCC 12228 are expressed but not exported into CFS [16]. Therefore, it is necessary to extract cytoplasmic bacteriocin compounds from dried *S. epidermidis* by TCA/acetone precipitation method after lysis by SDS-containing buffer.

The extracted cytoplasmic bacteriocin compounds were tested for antimicrobial activity against *S. aureus* by the agar well diffusion method. The production of a clear zone of growth inhibition was noted. The inhibition zone of intra-cytoplasmic protein (IP) from *S. epidermidis* ATCC 12228 at pH 5.0 or 6.0 was more prominent than pH 7.4 against *S. aureus* or MRSA [Figure 2]. The skin surfaces pH is on average between 5.0 and 6.0 [20]. The production of cytoplasmic bacteriocin compounds at acid condition was enhanced when compared with neutral pH. The antimicrobial activity of cytoplasmic bacteriocin compounds was found to be stable after heating at 45 °C for 20 min but lost its activities heating to 100 °C for 20 min [Figure 3]. The minimum bactericidal concentration of cytoplasmic bacteriocin compounds was taken as the lowest concentration that did not allow *S. aureus* growth on the agar well inhibition assay [Figure 4]. Therefore, the minimum bactericidal concentration of cytoplasmic bacteriocin compounds was 250 μg/ml. The extracted cytoplasmic bacteriocin compounds revealed several diffuse bands of approximately 40–70 kDa by SDS-PAGE [Figure 5].

Based on bacteriocin primary structure, molecular mass, heat stability and molecular organization, bacteriocins are divided into three classes. Class I lantibiotics are thermostable peptides of molecular weight below 5 kDa. Class II non-lantibiotics are thermostable bacteriocins of molecular weight below 10 kDa. Class III includes thermolabile bacteriocins of molecular weight above 30 kDa. The majority of bacteriocin from *S. epidermidis* strains are thermostable class I lantibiotics [2,6,21,22]. It has been reported that the production of type I, II thermostable bacteriocins and phenol-soluble modulins (PSMδ; MW 3kDa) by *S. epidermidis* can selectively kill bacterial pathogens such as *S. aureus* [23,24]. Iwase et al. (2010) [25] also have shown that a subset of biofilm inhibitory *S. epidermidis* eliminates *S. aureus* by secreting 27 kDa serine protease (Esp), but non-inhibitory *S. epidermidis* (Esp-negative strain, ATCC12228) did not show the eliminating effect. At present, cytoplasmic protein export mechanisms suggest that some of the cytoplasmic proteins are released from dead cells which then remain attached and decorate the cell walls of bacteria, and the membrane vesicle (MV) type of nonclassical secretion mechanism is likely to be exploited by *S. epidermidis*, allowing the export of cytoplasmic proteins [16,26]. Our study contributes to do further extensive research on the secretion mechanisms of the cytoplasmic bacteriocins from *S. epidermidis*.

Putting together the extracted cytoplasmic bacteriocin compounds from the *S. epidermidis* ATCC12228 strain were shown to selectively inhibit the growth of *S. aureus*. The result demonstrated that the extracted thermolabile cytoplasmic bacteriocin compound profiles by SDS-PAGE exposed several diffuse bands of approximately 40–70 kDa. Thus, the cytoplasmic bacteriocin compounds have potential topical applications in the management of AD.

## 4. Materials and Methods

### 4.1. Strains and Growth Conditions

The Staphylococcal target strains used in this study were *S. aureus* ATCC25923 (ATCC, Rockville, MD, USA), *S. aureus* NCCP14780 (NCCP, Seoul, Korea), MRSA (ATCC33591), and the test strains were *S. epidermidis* (ATCC12228 and NCCP14768). Strain stocks were stored in freeze medium containing 30% (w/v) glycerol at −80 °C. An overnight starter culture of *Staphylococcus* was grown for 16 h in Tryptic Soy Broth (TSB; Becton, Dickinson and company, Sparks, MD, USA) at 37 °C with constant agitation (160 rpm) to be used as an inoculum for the growth experiments. To obtain antimicrobial substances, the overnight culture was diluted 1:100 in Luria broth (LB; LPS solution, Daejeon, Korea) and incubated at 37 °C with constant agitation (160 rpm) for 4–6 h [27].

### 4.2. Preparation of CFS from S. Epidermidis for Antimicrobial Potentiality

Cells were removed by centrifugation at 3500 rpm for 20 min and the supernatants were filtered with a membrane of pore size 0.45 μm, this was then referred to as CFS. The extracellular proteins in the supernatant were precipitated with 10% (w/v) TCA on ice at 4 °Covernight. The precipitates were harvested by centrifugation at 8000 x g for 20 min at 4 °C and washed with ice-cold acetone and dried at 37 °C. The dried protein pellets were stored at -20 °C until further use [28,29].

### 4.3. Extraction of cytoplasmic bacteriocin compounds from S. epidermidis for antimicrobial potentiality

After harvesting test strains, cells were washed twice with PBS. The cells were re-suspended in 500 μl of SDS Lysis Buffer (2% SDS, 0.375 M Tris pH 6.8, 3.4 M sucrose). The re-suspension mixture was thoroughly mixed and sonicated on ice (10 min, 30s running, 10s pause, 40% amplitude) using sonicator (Vibra cell; Sonics & Materials Inc, Newtown, CT, USA). Cell debris was removed by centrifugation at 14,000 rpm (Hanil Microcentrifuge; Hanil Scientific Inc, Kimpo, Korea) for 30 min. The supernatant containing proteins were precipitated with 10% (w/v) TCA on ice at 4 °C overnight. Finally, the precipitates were collected by microcentrifugation for 30 min at 4 °C and washed three times with ice-cold acetone and dried at 37 °C [30]. Total cytoplasmic protein concentration was determined by the Bradford Assay kit (BioRad, Hercules, CA, USA). The dried protein pellets were stored at -20 °C until further use.

### 4.4. Stability Tests of Cytoplasmic Bacteriocin Compounds from S. epidermidis

The susceptibility of partially purified cytoplasmic bacteriocin compounds to temperature was investigated through the agar well diffusion assay. To determine temperature stability, the purified bacteriocin compound was subjected to 45 °C or 100 °C for 20 minutes. Antimicrobial activity was then determined by carrying out the agar well diffusion assay with *S. aureus* as the bacterial indicator.

### 4.5. Antimicrobial Activity of Bacteriocins by the Agar Well Diffusion Assay

LB agar plates (1.5% agar) were overlaid with 4 ml LB soft agar (1%) having 100 µl of Staphylococcal target strain. Wells of 6 mm diameters were bored in agar plates. Precipitated proteins were re-suspended with 150 μl of solubilization solution (25 mM ammonium bicarbonate and 10 mM dithiothreitol). Then, 80 μl of the re-suspended bacteriocin compound from a different condition was transferred into separate wells. The plates were incubated at 37 °C overnight and the growth inhibition zones were measured [31].

### 4.6. Determination of the Minimum Bactericidal Concentration by the Agar Well Diffusion Assay

Two-folded serial dilution of the cytoplasmic bacteriocin compounds was prepared (2 mg, 1 mg, 500 μg, 250 μg, and 125 μg/ ml in each well). Then, 80 μl of each bacteriocins was transferred into separate wells. The plates were incubated at 37 °C overnight and the growth inhibition zones were measured. Interpretation of the minimum bactericidal concentration was done as the lowest concentration of the cytoplasmic bacteriocin compounds that did not show any visible inhibition zones of *S. aureus* on the agar well.

### 4.7. Molecular Weight Estimation with Sodium Dodecyl Sulfate-Polyacrylamide Gel Electrophoresis (SDS-PAGE)

The molecular weight of partially purified bacteriocin compounds was determined by SDS-PAGE. The 30 µL sample was homogenized with 2% SDS and dithiothreitol (DTT) added to a little glycerin to increase the density. The sample was heated in boiling water for 5 min, then loaded onto gel along with molecular weight marker and mixed by using a Mini-Protean 4 cell system (Bio-Rad, Hercules, CA, USA), 100 V and 20 mA in the separation gel at pH 8.3. The gel was stained with 50 mL of Coomassie brilliant blue R-250 (0.1 M) for 4 hours. The decolorizing solution was decolorized for 12 h until the electrophoresis band was clear [32].

## Figures and Tables

**Figure 1 pathogens-09-00087-f001:**
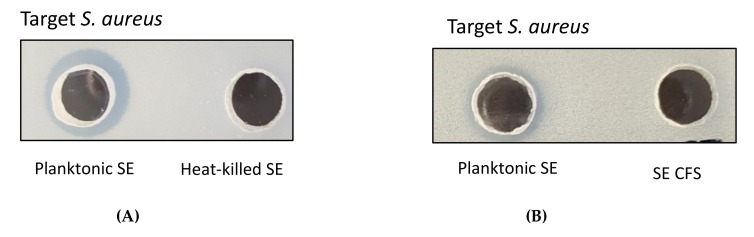
The planktonic *S. epidermidis* ATCC12228 (SE) showed antimicrobial activity against *S. aureus* ATCC 25923 on the agar well diffusion assay but heat-killed *S. epidermidis* (**A**) and cell-free supernatant (CFS) (**B**) from *S. epidermidis* did not show any activities.

**Figure 2 pathogens-09-00087-f002:**
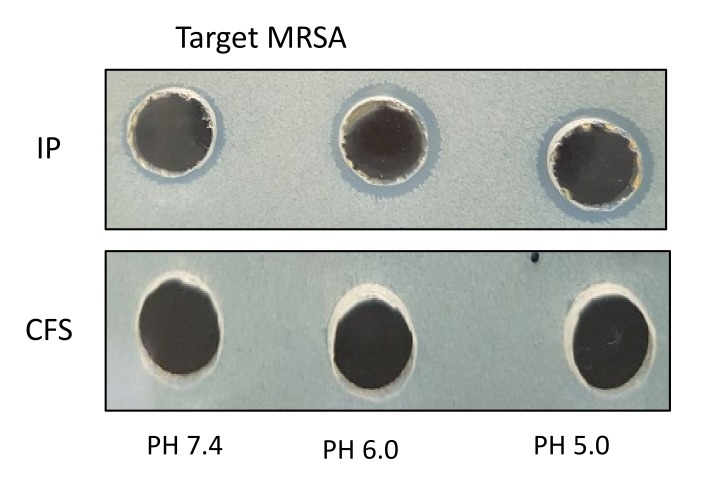
The effect of culture medium pH on the production of the cytoplasmic bacteriocin compounds. The inhibition zone of intra-cytoplasmic protein (IP) from *S. epidermidis* ATCC12228 at pH 5.0 or 6.0 was more prominent than pH 7.4 against *S. aureus* or methicillin-resistant *S. aureus* (MRSA). No antimicrobial activity was observed in the CFS with the overnight growth of *S. epidermidis*.

**Figure 3 pathogens-09-00087-f003:**
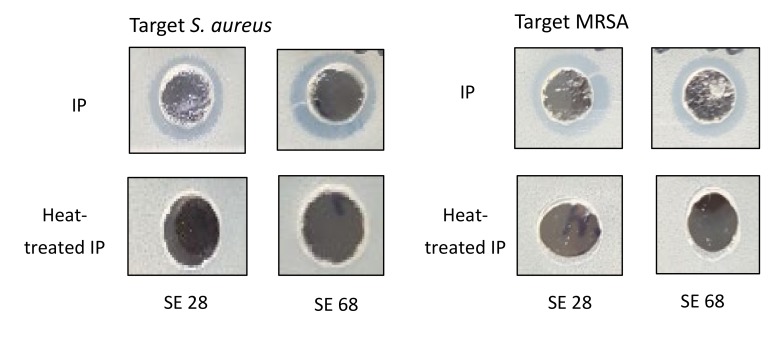
The antimicrobial activities of the intra-cytoplasmic protein (IP) from *S. epidermidis* ATCC12228 (SE 28) or NCCP14768 (SE 68) against *S. aureus* or MRSA. The antimicrobial activity was found to be stable after heating at 45 °C for 20 min but lost its actions heating to 100 °C for 20 min.

**Figure 4 pathogens-09-00087-f004:**
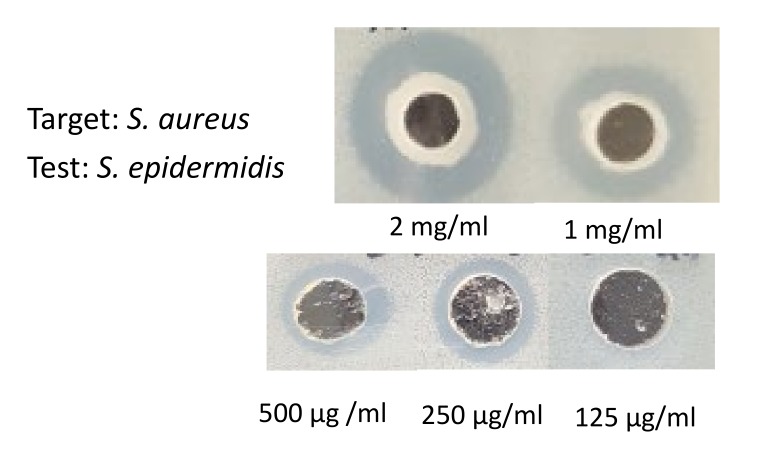
On the agar well inhibition assay, the concentration of cytoplasmic bacteriocin compounds from *S. epidermidis* ATCC12228 showing from 2 mg, 1 mg, 500 μg, 250 μg, and 125 μg/ ml in each well. Antimicrobial activity decreased with decrease in its concentration. There is no antimicrobial activity of cytoplasmic bacteriocin compounds from *S. epidermidis* ATCC12228 at concentration of 125 μg/ ml against *S. aureus* ATCC25923.

**Figure 5 pathogens-09-00087-f005:**
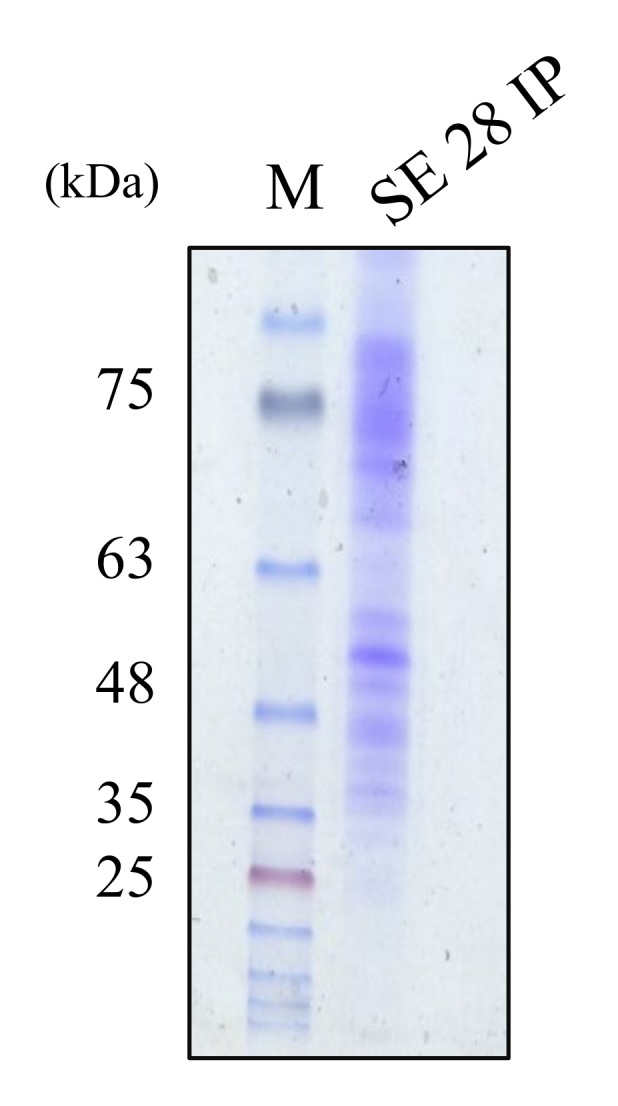
A bacteriocin compound profiles of SDS–PAGE of *S. epidermidis* ATCC12228, lane M: Molecular weight marker; Lane SE 28 IP: partially purified intra-cytoplasmic protein (IP) from *S. epidermidis* ATCC12228.

**Table 1 pathogens-09-00087-t001:** The diameters of antibacterial inhibitory zones in the live planktonic *S. epidermidis* (PCSE) and CFS from *S. epidermidis* ATCC12228 against bacterial indicators.

Bacterial indicators	PCSE	CFS
*S. aureus* (ATCC 25923) *S. aureus* (NCCP 14780)	11.1 ± 0.1 11.4 ± 0.2	― ―
MRSA (ATCC 33591)	10.6 ± 0.1	―
*S. epidermidis* (ATCC 12228)	―	―
*E. coli* (NCCP 14762)	―	―
*Salmonella* Typhimurium (NCCP 10438)	―	―

Note: CFS = cell-free supernatant; (-) = no inhibition activity; Results of inhibition zone (mm) including 6 mm well diameter is given as the mean value of the triplicate trials ± SD (standard deviation).

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
