# Peer review of "Novel Cytoplasmic Bacteriocin Compounds Derived from Staphylococcus epidermidis Selectively Kill Staphylococcus aureus, Including Methicillin-Resistant Staphylococcus aureus (MRSA)"

_pathogens, 2020, doi:10.3390/pathogens9020087_

Round 1
Reviewer 1 Report
In this article, the authors demonstrate that S. epidermis inhibited S. aureus growth on an agar plate. Cell free supernatant had no effect, and heated S. epidermis had no effect, demonstrating that the antimicrobial compound is a cytosolic protein. The results are intriguing and demonstrate a potential use for S. epidermis-derived proteins in the treatment of atopic dermatitis. However, I have some concerns with the data presentation and conclusions reached, as described below.
Major Concerns:
The data are presented primarily as images of inhibition zones. Each image has a hand-written notation about what it is. This would be improved greatly if it were typed.
The notations on the figures are sometimes confusing, particularly in Fig. 3. It is unclear to me what “ZP” and “H” mean.
Table 1 shows a quantification of the data in Figure 1. I would like to see this type of quantification for all of the results. Then each result can be shown as a table or a figure. The images could be included as Supporting Information. This would be particularly helpful for the data in Fig. 2 and Fig. 4. Graphs, with statistical analysis, would be much more useful than the current images.
The results in Fig.2 seem to be contradictory to those in Fig. 1. The zones of inhibition all seem to be smaller than what is shown in Fig. 1. At what pH was the bacteria used in Fig. 1 grown? Can the authors explain this difference?
In Section 2.4, the authors mention that “the concentration” ranges from 160-10 ug/well. It is unclear to me what this concentration is. Is this the bacterial mass? If so, how was it calculated? I could not find any information about this in the Methods section. Similarly, the authors then provide a corresponding concentration of the cytoplasmic bacteriocins of 2 mg/mL – 125 ug/mL. How was this calculated? This should be explained somewhere in the text.
Along those lines, in Section 4.6, the authors state that a serial dilution of the cytoplasmic bacteriocin compounds was prepared. How were these concentrations determined? Is this the total cytoplasmic protein concentration? If so, that should be stated clearly.
In Section 2.5, the authors show a SDS-PAGE of the S. epidermis cytoplasm. They talk about bands at 40, 50, and 70 kDa, which I agree seem to be the primary bands. But it is possible that the bacteriocins are a minor component. I think the authors have included this so that they can show that the compounds belong to Class III, but I am not convinced that there might not be a 5 kDa component that has been run off the gel. I think this gel does not lead to the conclusion they made. I recommend removing “Class III” from the title and removing this section, or conducting more extensive experiments to verify this conclusion.
The authors show growth inhibition on agar plates. Is the same behavior observed in planktonic growth?
My primary concern is how a compound inside S. epidermis cells would have any effect on the growth of a different bacterium. The authors have not addressed this point. Can they please provide a hypothesis of how this compound affects bacteria growing outside of the cell in which the bacteriocin is located?
Minor Concerns:
There are a number of places where bacterial names are not italicized, including the title and all figure legends.
There are several typos, including subject-verb disagreement
Line 82 …compounds…is…has… Line 131 …compounds…was tested… Legend of Fig 5, a space is missing between “PAGE” and “of” Line 212, “are” is missing between “bacteriocins” and “divided”.
The authors include a Section 4.8 about statistical analysis, but to my reading, there is no statistical analysis done on this work. I would like to see some analysis performed, as I mentioned above, but if the authors are not planning to conduct such analysis, this statement should be deleted.
The abstract would benefit from another round of proofreading.
Reviewer 2 Report
In general ok and interesting topic, however, the data thst are shown is very limited.
Author Response
Please see the attchment.

Reviewer 3 Report
In the manuscript entitled: “A Novel class III Cytoplasmic Bacteriocin Compounds Derived from Staphylococcus Epidermidis Selectively kill Staphylococcus Aureus, including Methicillin-Resistant Staphylococcus Aureus (MRSA)” the authors described the partial purification and characterization of the novel class III 80 thermolabile cytoplasmic bacteriocin compounds from cell extract of S. epidermidis strain.
The authors found that these bacteriocin selectively exhibited antimicrobial activity against S. aureus and 22 MRSA, presenting no active actions against S. epidermidis, E. coli, Salmonella Typhimurium. The SDS-23 PAGE analysis showed the molecular weight range of partially purified bacteriocin compounds from 24 40 kDa to 70 kDa.
The authors concluded underline that these cytoplasmic bacteriocin compounds would be a great potential means for S. aureus growth inhibition and the topical AD treatment.
Major comments:
In general, the idea and innovation of this study, regards the analysis of Cytoplasmic Bacteriocin Compounds Derived from is interesting, because the analysis of these compounds is validated but further studies on this topic could be an innovative issue in this field could be open an innovative matter of debate in literature by adding new information. Moreover, there are few reports in the literature that studied this interesting topic with this kind of study design.
The study was well conducted by the authors; However, there are some concerns to revise that are described below.
The introduction section resumes the existing knowledge regarding the important factor linked with some compounds associated with anti-inflammatory actions.
However, as the importance of the topic, the reviewer strongly recommends, before a further re-evaluation of the manuscript, to update the literature through read, discuss and cites in the references with great attention all of those recent interesting articles, that helps the authors to better introduce and discuss the aim of the study in light of the some other drug mediators and compounds with anti-inflammatory activity: 1) Matarese G, Currò M, Isola G, Caccamo D, Vecchio M, Giunta ML, Ramaglia L, Cordasco G, Williams RC, Ientile R. Transglutaminase 2 up-regulation is associated with RANKL/OPG pathway in cultured HPDL cells and THP-1-differentiated macrophages. Amino Acids. 2015 Nov;47(11):2447-55. 2) Isola G, Matarese M, Ramaglia L, Iorio-Siciliano V, Cordasco G, Matarese G. Efficacy of a drug composed of herbal extracts on postoperative discomfort after surgical removal of impacted mandibular third molar: a randomized, triple-blind, controlled clinical trial. Clin Oral Investig. 2019 May;23(5):2443-2453. 3) Bürgers R, Morsczeck C, Felthaus O, Gosau M, Beck HC, Reichert TE. Induced surface proteins of Staphylococcus [corrected] epidermidis adhering to titanium implant substrata. Clin Oral Investig. 2018 Sep;22(7):2663-2668.
The authors should be better specified, at the end of the introduction section, the rational of the study and the aim of the study with the null hypothesis. In the material and methods section, should better clarify how was performed the antimicrobial activity of bacteriocins by agar well diffusion assay and the SDS- PAGE analysis. Moreover, specify if data were normalized or not. Please specify if was used a test unit.
The discussion section appears well organized with the relevant paper that support the conclusions, even if the authors should better discuss the importance of dysbiosis in medical diseases. The conclusion should reinforce in light of the discussions.
In conclusion, I am sure that the authors are fine clinicians who achieve very nice results with their adopted protocol. However, this study, in my view, does not in its current form, satisfy a very high scientific requirement for publication in this journal and requests a revision before publication.
Minor Comments:
Abstract:
Better formulate the introduction section by better describing the background
Introduction:
Please refer to major commentsDiscussion
Please add a specific sentence that clarifies the results obtained in the first part of the discussion Page 6 last paragraph: Please reorganize this paragraph that is not clearAuthor Response
Please see the attachment.

Round 2
Reviewer 1 Report
I have reviewed the revised version of this manuscript, and I appreciate the changes the authors have made. However, a number of my original concerns remain, as described below.
I thank the authors for removing “class III” from the text. However, the final line of the abstract states that the molecular weight of the bacteriocins ranges from 40-70 kDa, but I am still unconvinced that the data show this.
The authors have not yet addressed my concern about what they are labeling as a concentration. They have modified the text to read “bacteriocin concentration,” but I am still unclear how this was determined. Is this the total cytoplasmic protein concentration? If so, it should be labeled as such, and the Methods section should include information about how this was determined (A280, Bradford assay, BCA assay, etc.).
In Figure 4, the formatting of “2 mg/ml” is different from the other labels, and as a result, the text covers some of the figure.
Reviewer 2 Report
The revised version of the manuscript looks ok, however, I need a more detailed author response in order to be able to decide its further progress. Please respond to all comments and suggestions.
Reviewer 3 Report
In the R1 version of the manuscript entitled: “A novel class III cytoplasmic bacteriocin compounds derived from Staphylococcus epidermidis selectively kill Staphylococcus aureus, including methicillin-resistant Staphylococcus aureus (MRSA)” the authors followed all the issues suggested by the reviewer. Though the changes based on the reviewer comments, almost of the criticisms were carefully analysed and solved. Even most of the original doubts still remain.
I have carefully evaluated all parts of the manuscript. I believe that the article, in this version, is now adequate for publication in this journal.
